# Access to unauthorized hepatitis C generics: Perception and knowledge of physicians, pharmacists, patients and non-healthcare professionals

**Amandine Garcia[1], Sascha Moore Boffi[2], Angèle Gayet-Ageron[3], Nathalie Vernaz** **[4]\***

**1** Faculty of Medicine, University of Geneva, Geneva, Switzerland, **2** Groupe Sida Genève, Geneva, Switzerland, **3** Medical Directorate, Division of Clinical Epidemiology, Geneva University Hospitals, Geneva, Switzerland, **4** Medical Directorate, Finance Directorate, Geneva University Hospitals, Geneva University, Geneva, Switzerland

\* nathalie.vernaz@hcuge.ch

**Data Availability Statement:** All relevant data are within the manuscript and its Supporting Information files.

## Abstract

### Objectives

Hepatitis C virus (HCV) causes both acute and chronic infection, which can potentially develop into cirrhosis and liver cancer. Healthcare systems are struggling to finance costly direct-acting antiviral agents through public funding for uninsured patients, despite the unprecedented high cure rates of these agents. Vulnerable populations are at higher risk of HCV infection. The personal importation scheme is based on the legal right to import any unauthorized generics for personal use. This study was designed to assess the knowledge and perceptions of stakeholders on unauthorized generics.

### Methods

We conducted an anonymous online survey based on the fictitious situation of a patient diagnosed with HCV who lacked mandatory health insurance and personal financial resources.

### Results

We obtained a sample of 781 respondents: 445 physicians, 77 pharmacists, 51 patients and 207 non-healthcare professionals. We found that only 36% and 58% of respondents believe that the quality and efficacy, respectively, of unauthorized generics are equivalent to their corresponding brand. An overwhelming majority (98%) favoured quality control upon arrival, and 31% felt they could recognize fraudulent websites. A total of 79% expressed support for financial assistance for vulnerable patients, and support among physicians was 83%.

### Conclusions

Overall, the limited knowledge of the efficacy and quality of unauthorized generics, despite evidence in peer-reviewed literature, contrasts with the overwhelmingly positive attitudes

**Funding:** The authors received no specific funding for this work.

**Competing interests:** The authors have declared that no competing interests exist.

toward financial assistance for personal import. This finding emphasizes the need for clearer information on imported generics and the potential safety provided by buyers' club schemes to complete the WHO agenda of eradicating viral hepatitis by 2030 within otherwise excluded vulnerable populations.

## Introduction

### Background

Chronic hepatitis C virus (HCV) infections affect an estimated 71 million people.[1] Given the global burden of HCV and the recent introduction to the market of new and effective direct-acting antivirals (DAAs), the WHO aims to eliminate viral hepatitis as a global health threat by 2030.[1, 2] To achieve this ambitious agenda, several strategies need to be implemented to reduce costs, diminish the risk of financial hardship for those in need of these services, and encourage solutions for more sustainable and equitable financing.[3] The high cost of DAAs is a major burden on health systems, even for high-income countries such as Switzerland. As a cost-saving measure, the Swiss Federal Office of Public Health (FOPH) limited insurance coverage for DAA treatment to only those diagnosed with an advanced form of the disease. It was years before drug prices decreased sufficiently for these access limitations to be removed by the FOPH in 2017, thereby allowing insurance coverage for all insured persons in Switzerland. [4, 5] However, access issues remain for vulnerable individuals not covered by mandatory healthcare insurance, people living in prisons (PLP), people who inject drugs (PWID) and migrants, for whom HCV prevalence remains high and access to treatment is limited.[5–7]

### The international agreement on Trade-Related Aspects of Intellectual Property Rights (TRIPS): A step toward equitable financing in low- and middle-income countries

The history of the global human immunodeficiency virus (HIV) epidemic provides us with real life lessons in achieving drug price reductions and improving global access to treatment. Applying similar strategies to the global hepatitis C epidemic may help in reaching comparatively affordable access to DAAs.[8–12] By leveraging TRIPS flexibilities, such as compulsory licenses, as well as voluntary licensing schemes such as the Medicine Patent Pool, low- and middle-income countries have achieved notable success in fostering a robust generic market that provides antiretroviral therapy to 93% of people living with HIV at a fraction of the cost of originator drugs.[11–14]

### Buyers' clubs: A step toward equitable financing in high-income countries

Even after the FOPH dropped the limitation on insurance coverage for DAAs in 2017, giving the insured population nominal unlimited access to DAAs, a significant number of vulnerable or uninsured individuals remained excluded from affordable access to these antivirals.[5, 15] Swiss law allows any individual to import ready-to-use unauthorized medicines for their personal use provided the imported quantity is small and is for their personal consumption.[16, 17] With the aim of protecting the swiss market and patients from counterfeit or substandard medicines, the regulating authority, the Swiss Agency for therapeutic products or Swissmedic, limits personal imports to a maximum quantity equivalent to one month of therapy at a standard dosage as indicated, for example, in the patient information leaflet.[18]

Buyers' clubs, as implemented in some countries, aim to help patients navigate the long and difficult process of personally importing unauthorized drugs, working as mediators in the drug supply chain and assisting patients in sourcing unauthorized generic drugs that are safe and effective.

Although most people importing medicine for their personal use appear to do so without requiring assistance, various institutions and associations in Switzerland, such as the Swiss Hepatitis C Association (www.hepc.ch) (SHCA), the University of Geneva Hospitals (HUG) and Groupe Sida Genève (www.groupesida.ch) (GSG), offer technical, medical and or financial support to patients importing generic DAAs manufactured under license in India or Bangladesh through the FixHepC Buyers' club (fixhepC.com), based in Australia.[19] The FixHepC Buyers' club guarantees the authenticity of the shipped generics and offers the possibility for the patient to be included in a scientific study aiming to establish the effectiveness of those licensed products compared to the originator products.[19, 20] Finally, FixHepC organizes the shipping of the generic DAAs to the patient.[19, 20]. All buyers' clubs assist patients in avoiding counterfeits, but HUG and SHCA ensure proper medical monitoring once the treatment is initiated. GSG offers financial support to patients who would otherwise not be able to afford even the lower priced generics. HUG will also perform quality control measures, such as high-performance liquid chromatography (HPLC-UV), for DAA batches received by their patients.[5] Although the possibilities described above may provide solutions that could benefit vulnerable populations in high-income countries, the acceptability of such schemes within stakeholder populations remains an open question. Concerns about the quality and efficacy of generic or unauthorized generics have sparked intense debate worldwide, and opinions vary widely within pharmaceutical, medical and public sectors as well as among patient groups.[21–24] The efficacy and quality of medicines purchased abroad as well as the reliability and trustworthiness of online pharmacies not subject to Swiss or even EU regulatory oversight are generally regarded as low to nonexistent by both authorities and consumer advocates in Switzerland.[25–28]

However, very little is known about the knowledge and perceptions of the Swiss population regarding unauthorized generics or the provision of financial support for personal importation schemes for vulnerable persons. To gain better insight into these issues, we surveyed a sample of some principal stakeholders.

## Methods

### Survey

The questionnaire was based on the fictitious case of a 28-year-old woman diagnosed with hepatitis C who has no access to insurance and does not have the financial resources necessary to personally cover the costs (see Text A in S1 File). The fictitious case indicates that the patient's attending physician suggested that she contact a buyers' club that might assist her in importing unauthorized generics from India or Egypt and help her find financial support. The case description highlighted both the price of the drugs authorized in Switzerland (approximately 30,000 CHF) and the price of the generics available overseas (approximately 1,000 CHF) and was followed by 21 questions about the participants' knowledge and perceptions regarding imported generics and the related risks. We chose to focus our attention on five specific questions: 1) the efficacy of generics imported from India or Egypt, 2) the safety of these generics, 3) quality control at entrance, 4) the respondent's ability to differentiate a trustworthy website from a fraudulent one and 5) the respondent's opinion regarding whether to financially support a patient importing generic drugs for personal use. These five variables were defined as primary outcomes and are described separately. Further questions, such as "*Have*

*you ever bought medicines through the Internet*?", "*Before this survey, had you ever heard about an association that can help import generics*?" or "*Do you know of a buyers' club*?" were asked, and the responses were treated as secondary outcomes (see Table A in S1 File). Participants were chosen from five of the principal stakeholder groups in Switzerland: physicians, pharmacists, patients, non-healthcare professionals and politicians. The survey was distributed to members of the five groups through separate channels. For the physician's group, we sent an email to all registered physicians at the HUG and Lausanne CHUV. For the pharmacist group, we sent an email to 15 HUG pharmacists and 300 members of the Pharmacist's Society of Geneva (https://www.pharmageneve.swiss). For the patient group, we sent an email to the 100 members of the HUG Patients' Association. The group of non-healthcare professionals was composed mostly of students from various programmes, including the medical school at the University of Geneva, who were contacted through social media (Facebook), as well as members and beneficiaries of the GSG, who were contacted through the association's newsletter. The participating politicians were members of the Swiss parliament who were contacted through their published parliamentary email address. Participants were invited to participate using www.sondageonline.com; they could log in to the survey through a specific link provided to each group. Participation in our survey was voluntary. By completing the survey, all respondents implicitly consented to the use of their data in a scientific study. Because no personal information was collected, no email addresses or IP addresses were logged and all data was collected and analysed anonymously thus guaranteeing the anonymity of all respondents, this study is considered to be exempt from formal institutional review.[29] The questionnaire was opened for four months, from January 2, 2019, to April 30, 2019. All participants who had answered at least one question of interest were included. Because only 14 participants fulfilled the inclusion criteria, we decided to exclude the politicians' group.

## Statistical analysis

Different comparisons among the groups of respondents and some of the respondents' characteristics were performed using the chi-square test or Fisher's exact test, depending on the conditions to which the statistical test was applied. We then assessed the five primary outcomes using the original 4-point scale (ordinal format, "I do not agree at all", "I do not agree", "I agree", "I fully agree"). Five ordered logistic models were performed to estimate the association between each dependent variable (efficacy, quality, and quality control of Indian or Egyptian generics; perceptions regarding financial support) and the respondent group. All models were adjusted for the categories of age (20–29, 30–39, 40–54, and $> = 55$ years), gender, nationality (Swiss, European nationalities other than Swiss, and others) and level of education (university or higher versus primary/mandatory school and other). All variables were assessed in univariate analyses and then forced into the multivariate models. Regarding the outcome assessing the respondents' reliance on and confidence in the identification of websites, we performed an unconditional logistic regression model because the dependent variable was dichotomous (reliance "yes" *vs.* "no"). We also forced all variables into the model. All associations are presented as odds ratios, 95% confidence intervals (95%CIs) and p-values. All analyses were performed using intercooled STATA version 15.0 (StatCorp., College Station, TX, USA). Statistical significance was defined as p<0.05 (two-sided).

## Results

Over the study period, we obtained a sample of 781 respondents: 445 physicians, 77 pharmacists, 51 patients and 207 non-healthcare professionals; all participants had answered at least one question of interest (see Table B in S1 File). Because only 14 participants fulfilled the

inclusion criteria, we decided to exclude the politicians' group. Women, people under forty years of age and higher education levels were overrepresented in the sample (Table 1). The majority of the respondents were Swiss (77.3%), and more than half of the participants were physicians (57.0%). We compared the characteristics of the four groups (physicians, pharmacists, patients and non-healthcare professionals) (Table 1). We found significant differences regarding gender (more women among the pharmacists and non-healthcare professionals than among the doctors or patients), age categories (the non-healthcare professionals were younger than the patients, pharmacists and doctors), nationalities (more respondents were Swiss among the pharmacists and non-healthcare professionals than among the doctors and patients) and education (Table 1).

Fig 1 shows that 58% (435/749) of the respondents strongly agreed or agreed that the efficacy of an imported generic is equivalent to the corresponding Swiss medicine, and 36% (271/749) strongly agreed or agreed that the quality of an imported generic is equivalent to the corresponding brand (see Table B in S1 File). The majority (733/751, 98%) answered that they would strongly agree or agree to quality control upon arrival, and 79% (586/735) strongly agreed or agreed that there is a need to financially support this initiative; support among physicians was 83%. Finally, 31% (202/656) of the respondents declared that they could recognize a fraudulent website.

In the univariate analysis (see Table C in S1 File), we found that the physician and non-healthcare professional groups were consistently associated with all four primary outcomes: compared to pharmacists, physicians and non-healthcare professional were, respectively, 3.40 and 5.27 times more willing to agree regarding the efficacy of imported generics, 3.36 and 4.52 times more willing to agree about their quality, 2.40 and 2.75 times more willing to agree about the need for quality control, and 3.16 and 3.47 times more willing to agree about the

**Table 1. Respondent characteristics by survey group.**

| Variables | Physicians (n = 445) | Pharmacists (n = 77) | Patients (n = 51) | Non-healthcare professionals (n = 207) | Total (n = 781) | p-value |
|---|---|---|---|---|---|---|
| **Gender, n (%) (2 missing)** | | | | | | <0.001[1] |
| Male | 219 (49.3) | 23 (29.9) | 22 (43.1) | 46 (22.2) | 310(39.8) | |
| Female | 225 (50.7) | 54 (70.1) | 29 (56.9 | 161 (77.8) | 469(60.2) | |
| **Age in years, n (%)** | | | | | | <0.001[1] |
| 20–29 | 81 (18.2) | 16 (20.8) | 2 (3.9) | 156 (75.0) | 255(32.7) | |
| 30–39 | 209 (47.0) | 20 (26.0) | 7 (13.7) | 11 (5.3) | 247(31.6) | |
| 40–54 | 104 (23.4) | 27 (35.1) | 21 (41.2) | 20 (9.6) | 172 (22.0) | |
| > = 55 | 51 (11.5) | 14 (18.2) | 21 (41.2) | 21 (10.1) | 107(13.7) | |
| **Nationality, n (%)** | | | | | | <0.001[2] |
| Swiss | 321 (72.1) | 64 (83.1) | 36 (70.6) | 183 (88.0) | 604(77.3) | |
| In European union | 112 (25.2) | 11 (14.3) | 8 (15.7) | 21 (10.1) | 152(19.5) | |
| Other | 12 (2.7) | 2 (2.6) | 7 (13.7) | 4 (1.9) | 25 (3.2) | |
| **Education, n (%)** | | | | | | <0.001[2] |
| Primary | 2 (0.5) | 1 (1.3) | 8 (15.7) | 3 (1.4) | 14 (1.8) | |
| Postmandatory | 0 (0) | 0 (0) | 7 (13.7) | 6 (2.9) | 13 (1.7) | |
| University/high level | 442 (99.3) | 75 (97.4) | 36 (70.6) | 194 (93.3) | 747(95.7) | |
| Other | 1 (0.2) | 1 (1.3) | 0 (0) | 5 (2.4) | 7 (0.9) | |

1 Comparisons were performed using the chi-square test;

2 Comparisons were performed using Fisher's exact test.

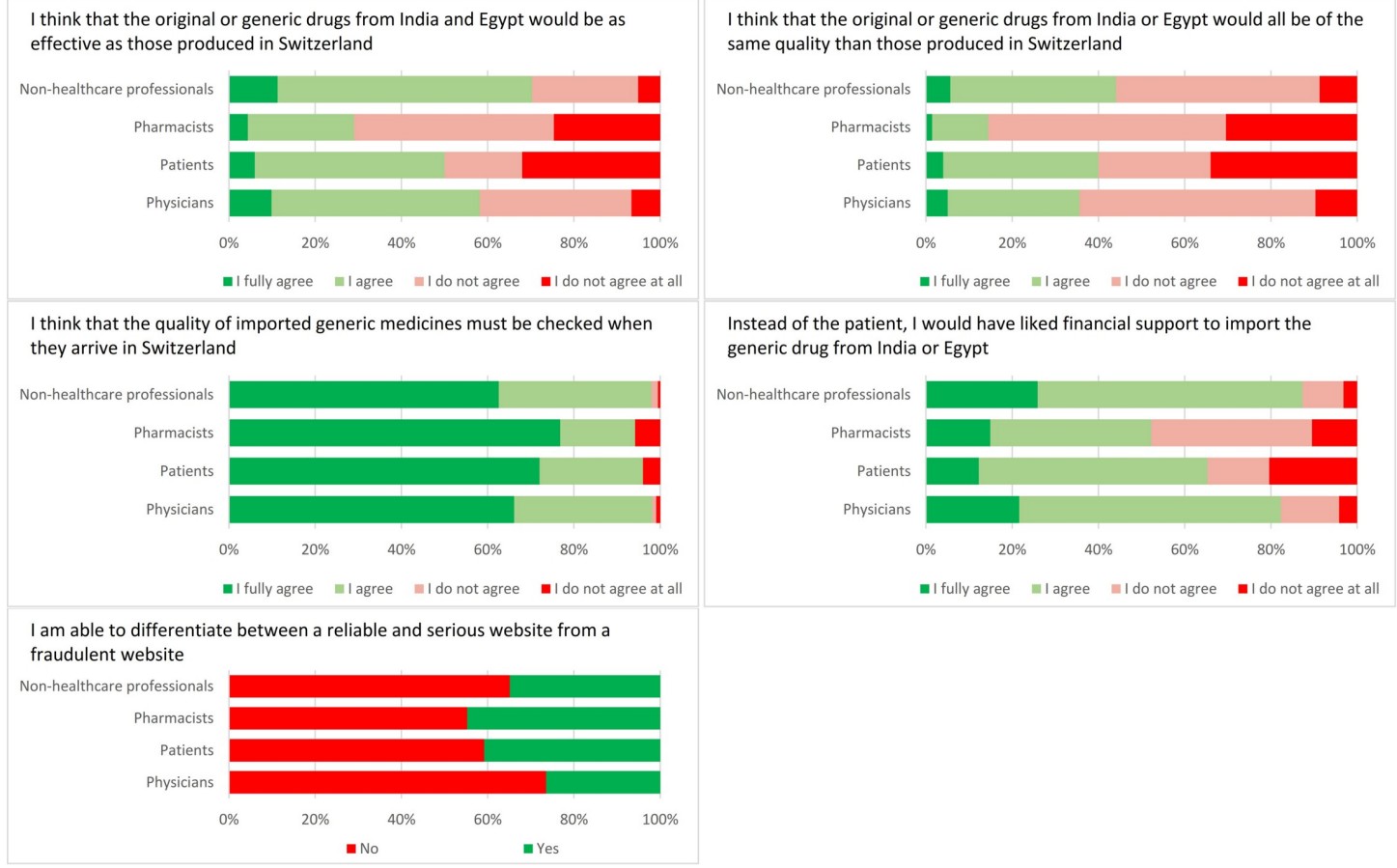

**Fig 1. Five key questions regarding the knowledge and perception of respondents regarding efficacy, quality, quality control upon arrival, Internet purchases, and financing.**

need to financially support this initiative. Physicians felt less capable of identifying a trustworthy website compared to pharmacists (OR = 0.44; 95%CI: 0.26–0.75, p = 0.002).

Younger age categories were also associated with higher agreement with the question regarding the efficacy of imported generics (OR = 2.31; 95%CI: 1.48–3.59, p<0.001 in the 20–29 years age category compared to > = 55 years age category) and the identification of a valuable website (OR = 1.83; 95%CI: 1.02–3.28, p = 0.043; OR = 1.83; 95%CI: 1.14–2.93, p = 0.012; OR = 2.47; 95%CI: 1.51–4.04, p<0.001, respectively for the 20–29, 30–39, and 40–54 years age categories compared to > = 55 years age category). A high-level education was associated with higher agreement to the questions regarding the need for quality control and the need to financially support this initiative compared to a middle-level education (OR = 3.82; 95%CI: 1.52–9.59, p = 0.004 and OR = 2.11; 95%CI: 2.11; 95%CI: 1.05–4.24, p = 0.035, respectively). Other nationalities agreed less than Swiss respondents with the need to financially support this initiative (OR = 0.29, 95%CI: 0.13–0.66, p = 0.003).

In multivariate analyses, we confirmed the results of the univariate analyses (Table 2). Medical doctors, patients and non-healthcare professionals had all a higher agreement with the statement that Indian/Egyptian generics had the same efficacy as Swiss drugs. Other factors were associated with the outcome, but associations varied depending on the outcome assessed.

Finally, we looked at secondary outcomes and found that only 9% of physicians (n = 430), 2% of patients (n = 46) and 3% of pharmacists (n = 67) had heard of a buyers' club.

**Table 2. Factors associated with the five primary outcomes (multivariate analyses).**

| | Efficacy | | Quality | | Quality control | | Finance initiative | | Fraudulent website | |
|---|---|---|---|---|---|---|---|---|---|---|
| Variables | OR (95%CI) | p-value | OR (95%CI) | p-value | OR (95%CI) | p-value | OR (95%CI) | p-value | OR (95%CI) | p-value |
| *Gender* | | | | | | | | | | |
| Male | 1 | 0.758 | 1 | 0.518 | 1 | 0.934 | 1 | 0.272 | 1 | 0.088 |
| Female | 1.05 | | 0.91 | | 1.01 | | 1.19 | | 0.73 | |
| | (0.78–1.40) | | (0.68–1.22) | | (0.73–1.40) | | (0.87–1.61) | | (0.51–1.05) | |
| *Age in years* | | *0.081* | | *0.159* | | *0.847* | | *0.259* | | *0.001* |
| > = 55 | 1 | - | 1 | - | 1 | - | 1 | - | 1 | |
| 40–54 | 1.23 | 0.393 | 0.98 | 0.928 | 1.1 | 0.736 | 1.03 | 0.893 | **3.03** | **<0.001** |
| | (0.77–1.97) | | (0.60–1.60) | | (0.64–1.90) | | (0.67–1.59) | | **(1.74–5.28)** | |
| 30–39 | 1.47 | 0.103 | 1.44 | 0.132 | 1.15 | 0.612 | 1.44 | 0.078 | **2.18** | **0.002** |
| | (0.92–2.34) | | (0.90–2.31) | | (0.67–1.96) | | (0.96–2.15) | | **(1.32–3.59)** | |
| 20–29 | **1.83** | **0.016** | 1.01 | 0.97 | 0.94 | 0.838 | 1.19 | 0.901 | 1.76 | 0.068 |
| | **(1.12–2.99)** | | (0.62–1.65) | | (0.54–1.65) | | (0.62–1.72) | | (0.96–3.24) | |
| *Nationality* | | *0.271* | | *0.781* | | | | *0.011* | | *0.298* |
| Swiss | 1 | - | 1 | - | 1 | 0.791 | 1 | - | 1 | |
| EU | 0.86 | 0.425 | 1.06 | 0.737 | 0.89 | 0.577 | 1.19 | 0.38 | 1.31 | 0.201 |
| | (0.60–1.24) | | (0.74–1.52) | | (0.60–1.33) | | (0.81–1.74) | | (0.87–1.99) | |
| Other | 0.54 | 0.139 | 0.78 | 0.562 | 0.81 | 0.654 | **0.31** | **0.006** | 1.62 | 0.3 |
| | (0.24–1.22) | | (0.34–1.80) | | (0.32–2.06) | | **(0.13–0.71)** | | (0.65–4.07) | |
| *Education* | | | | | | | | | | |
| Middle level | 1 | 0.552 | 1 | 0.311 | 1 | **0.005** | 1 | 0.11 | 1 | 0.346 |
| High level | 1.24 | | 1.47 | | **4.01** | | 1.83 | | 1.6 | |
| | (0.61–2.56) | | (0.70–3.11) | | **(1.53–10.53)** | | (0.87–3.82) | | (0.60–4.26) | |
| *Group* | | | | *<0.001* | | *0.013* | | *<0.001* | | *0.003* |
| Pharmacists | 1 | <0.001 | 1 | - | 1 | - | 1 | - | **1** | |
| Physicians | **3.38** | **<0.00** | **3.00** | **<0.001** | 2.37 | **0.00** | **3.00** | **<0.001** | **0.35** | *<0.001* |
| | **(2.06–5.56)** | | **(1.80–5.02)** | | **(1.27–4.41)** | | **(1.84–4.92)** | | **(0.20–0.61)** | |
| Patients | **2.20** | **0.041** | **2.40** | **0.031** | 2.17 | 0.085 | **2.18** | **0.042** | 0.99 | 0.982 |
| | **(1.03–4.70)** | | **(1.09–5.31)** | | (0.90–5.26) | | **(1.03–4.61)** | | (0.44–2.23) | |
| Non-healthcare professionals | **4.39** | **<0.001** | **4.98** | **<0.001** | **3.16** | **0.001** | **3.79** | **<0.001** | **0.46** | **0.028** |
| | **(2.49–7.72)** | | **(2.79–8.89)** | | **(1.59–6.31)** | | **(2.14–6.70)** | | **(0.23–0.92)** | |

## Discussion

### Efficacy and quality of imported unauthorized generics

To the best of our knowledge, this study is the first to analyse perceptions among physicians, pharmacists, patients and non-healthcare professionals towards imported unauthorized generics from India or Egypt in a high-income country. We found that only 36% and 58% of respondents believe that the quality and efficacy, respectively, of unauthorized generics are equivalent to that of their corresponding brand; pharmacists were least likely to agree with these statements. This distrust of imported unauthorized generics might be partly explained by poor knowledge of the existence of approval processes for generic drugs. Trust issues surrounding the use of unauthorized imported generic drugs may also be linked to the fear of substandard drugs entering the supply chain and being prescribed to patients.[30] Even if the generic approval process does not require the same level of clinical testing as for a brand drug, quality is a key issue in the registration process

worldwide.[24, 31] The quality of finished pharmaceutical products (FPP) and active pharmaceutical ingredients (API) evolves in a strongly regulated market. There are three measures of similarity that the generic drug and unauthorized imported generics must meet to be considered clinically equivalent to their brand-name counterpart during the drug application process. First, generic drugs must be chemically equivalent, containing the same active ingredient, route of administration, dosage form, and strength as their brand-name counterpart. The second mandatory principle is bioequivalence, meaning that the drug has the same rate and extent of absorption as the brand drug, which is a key issue in the registration process worldwide.[32] A bioequivalence study is usually conducted using a randomized, two-period, two-treatment, crossover design that allows comparison within individual healthy volunteers.[33] Hill and colleagues compared the bioequivalence of sofosbuvir and daclatasvir from seven pharmaceutical companies worldwide and concluded that they were all bioequivalent.[20] Finally, generic drugs must show label equivalence, which means that the indications are the same as the originator product. However, the literature largely describes consumer and healthcare professionals' preferences for brand-name drugs over generics. Opinions about generics have always differed, and it often becomes difficult to convince patients and caregivers of the equivalence of these drugs, despite their major cost-saving contributions.[22–24, 34] Several studies have shown that not only are patients reluctant to use generics, but medical doctors also seem to not be always convinced of their equivalency.[22–24, 34] It is essential to address these issues and to isolate the cause of such reluctance if we aim to give a more important role to imported generics in the future. The broad adoption of generic drugs following the loss of exclusivity for the originator drug and the import of generics are essential to achieving sustainable financing of public health systems.[31, 35–37]

The WHO has instituted procedures that, once the requirements of safety, quality and efficacy are met, will lead to WHO prequalification of generic drugs.[38] The WHO prequalification programme established in 1987 guarantees that prequalified drugs produced in or for developing countries meet globally recognized standards of quality, safety and efficacy. This programme conducts inspections of pharmaceutical facilities to evaluate them in terms of good manufacturing processes and also performs bioequivalence studies. The United States Food and Drug Administration (FDA) has instituted a similar pathway to generic drug approval, known as the Abbreviated New Drug Application (ANDA). This process guarantees that drugs approved under the ANDA meet the same standards of quality, safety and efficacy as drugs marketed in the United States. Just as with the WHO prequalification programme, the FDA will inspect facilities manufacturing generic drugs.[39–41] Indian generic manufacturers are significant players in the global generic drug market; for instance, India supplies more than 80% of drugs to treat AIDS in developing countries and might also play a crucial role in the fight against HCV.[42, 43] Notably, Mylan Pharma, an Indian pharmaceutical company, introduced the first generic in the Swiss market in December 2018.[44]

The respondents were slightly more confident in the efficacy of imported generics (58%) compared to their corresponding Swiss brand-name drug, and the non-healthcare professionals were the most confident group. Efficacy is addressed by measuring the sustained viral response (SVR) twelve weeks after the end of the treatment. With regard to generic DAAs, many trials have been undertaken to demonstrate their efficacy and prove their equivalence. [37, 45–49]

The scepticism about the efficacy of imported generics among all groups in this study contrasts sharply with the literature, which demonstrates through several analyses the equivalent quality and corresponding efficacy of these imported unauthorized generics.[19, 20, 45–49] Abozeid and colleagues found that sofosbuvir-daclatasvir and sofosbuvir-ledipasvir had

comparable clinical efficacy when either the generic or the brand formulation was used.[46] Liu and colleagues conducted a similar study and found a comparable high SVR rate in both HIV positive as well as HIV negative patients treated with generics.[49] We could explain these differences by the study population selected in the present survey, which was probably more educated and sensitive to the topic than the populations in published studies.

## Internet purchasing

Once it is proven that these generics are of the same quality and efficacy as the originals, several obstacles remain, particularly in relation to the possibility of purchasing these generics through the Internet as well as personal importation schemes. Indeed, these new ways of obtaining treatments raise a certain amount of risks as counterfeits circulate easily, and there is significant difficulty distinguishing a fraudulent website from a trustworthy website. A rigorous chain of controls through buyers' clubs, organizations such as Swissmedic and targeted medical surveillance once the treatment is initiated need to be implemented if we aim to use unauthorized generic DAAs safely.

Access to DAAs through Internet purchasing and the legal right to import unauthorized generics might help to increase treatment access and HCV treatment. However, the pharmaceutical supply chain becomes a key issue in providing protection from falsified medicine as counterfeits circulate easily on the Internet market.[50] We are therefore faced with the dilemma of balancing the escalation of risks from buying from unverified websites with the improvement of easy worldwide access to treatment, underlining once more the need to find a sustainable solution to assure patient safety while providing alternative access to treatment. Efforts made by Swissmedic to warn people of the dangers of Internet purchases are crucial, and strict controls must be implemented.[18, 51] We found that 31% of the respondents declared that they are able to recognize a fraudulent website; pharmacists were the most confident group, which might reflect their knowledge of which online medicine retailers to rely on because purchasing through the Internet is part of their work. Our study showed that a vast majority of the respondents (98%), regardless of group, agreed to the need to reinforce control of the drug supply chain, especially upon the drugs' arrival in Switzerland. Quality controls upon arrival require help from a larger laboratory and induce extra costs that should be taken into consideration unless low-cost technologies are used.[52] Wang and colleagues tested (by liquid chromatography) pre-exposure prophylaxis with tenofovir disoproxil fumarate and emtricitabine bought through seven different pharmacies online and from six Indian manufacturers and found that they contained the claimed amount of API.[53]

## Buyers' clubs

Buyers' clubs were not very well known by any of the stakeholder populations surveyed. Buyers' clubs are private organizations that can assist patients in accessing generic medicines that meet established standards of quality and efficacy by leveraging both national regulations and international networks of patients, stakeholders and organizations to ensure that sources of generic drugs for personal use are reliable and only provide safe and effective medicines. Some may even have the resources necessary to apply quality control processes to guarantee drug safety.[5] The selection of well-known generic manufacturers and the use of commercial intermediaries are some of the means by which supply chain integrity can be better ensured. Quality control testing of medication bought online by patients, such as running HPLC-UV on drug samples, will provide even greater confidence in the safety and quality of imported unauthorized generics.[5] Insuring safety and quality within a personal import scheme without the security of an unbroken and traceable drug supply chain from drug approval to quality

control at delivery represents a significant challenge to all stakeholders, most notably to patients and buyer's clubs. This challenge will need to be addressed to guarantee the safest treatment possible. Adding discussion of the quality and efficacy of generic drugs, including topics related to the potential benefits and drawbacks of personal importation schemes and buyers' clubs, to academic curriculums may provide an attractive means to build trust in generics among tomorrow's healthcare professionals and alternative solutions to access issues for vulnerable populations.

## Strength and limitations

This study has several limitations. The first is the fictitious case on which the survey is based, which may not be sufficiently representative of the population of patients unable to access care. Additionally, the patient group was exclusively composed of people who were accessing treatment and care in a university hospital, which may have introduced bias regarding accessibility difficulties and cost issues. It is further unknown whether any of these patients could have benefited from a buyer's club-type scheme. Another limitation concerns the group of non-healthcare professionals, which may not have been particularly representative of the overall population as it contains people who use social media and are contacts of one of the study investigators, notably those who were students at the University of Geneva. Due to time constraints and the study horizon, we did not conduct a survey of the general population. An interesting future analysis would be to establish to what extent the general population is familiar with these concepts and what measures could be put in place to increase their knowledge and understanding. For reasons mostly linked to language constraints and the logistics and costs involved in community interpretation, application of the present survey was not deemed feasible within local vulnerable populations. A more thorough survey of the perceptions and attitudes among these populations would be of great interest and should be considered a priority. Finally, the participants were given the opportunity to add personal comments on the survey, and the absence of "*I don't know*" as an answer option came up in these comments. The decision to not add "*I don't know*" as a possible answer was made in response to concerns that the frequency of this response would be too high. Finally, the survey was limited to the cantons of Geneva and Vaud; however, we presume that knowledge and perception are relatively homogenous within Switzerland, and we believe the results observed in the study are of value for Switzerland in general. Whether the observed knowledge and perception are comparable in other European countries should be further explored.

## Conclusion

This study showed that a large majority of respondents mostly agree on providing financial help for those in need and do not object to the use of imported generics. Indeed, the complexity of the drug supply chain and the possibility of counterfeits circulating emphasizes the need for affordable quality and safety controls at different levels, from drug production to purchase by the patient. However, a lack of knowledge regarding the efficacy and quality of imported unauthorized generics has been highlighted, and awareness should be increased among healthcare professionals to convince them that the use of generics imported safely with the help of a buyers' club could be a key strategy for HCV eradication and seems highly promising as a means of promoting access to medicine for all.

## Supporting information

**S1 File. Text A.: Fictitious case.** A 28-year-old woman diagnosed with hepatitis C who has no access to insurance and does not have the financial resources necessary to personally cover the costs. **Table A: Survey.** The survey includes 21 questions about the participants' knowledge

and perceptions regarding imported generics and the related risks. **Table B: Survey results. Table C: Univariate analysis.**
(DOCX)

## Acknowledgments

The authors thank Prof Alexandra Calmy, Prof Francesco Negro and Dr Yves Jackson for their valuable comments on the fictitious case and survey and Prof Arnaud Perrier for his valuable comments on the manuscript. The authors thank Pharmageneve association, Groupe sida Geneve, the HUG Patients' Association and all the respondents for participating in our survey.

## Author Contributions

**Conceptualization:** Amandine Garcia, Sascha Moore Boffi, Nathalie Vernaz.

**Data curation:** Amandine Garcia, Nathalie Vernaz.

**Formal analysis:** Amandine Garcia, Angèle Gayet-Ageron, Nathalie Vernaz.

**Investigation:** Amandine Garcia, Sascha Moore Boffi, Nathalie Vernaz.

**Methodology:** Amandine Garcia, Sascha Moore Boffi, Angèle Gayet-Ageron, Nathalie Vernaz.

**Project administration:** Amandine Garcia, Nathalie Vernaz.

**Software:** Amandine Garcia.

**Supervision:** Nathalie Vernaz.

**Validation:** Amandine Garcia, Sascha Moore Boffi, Nathalie Vernaz.

**Visualization:** Amandine Garcia, Sascha Moore Boffi, Nathalie Vernaz.

**Writing – original draft:** Amandine Garcia.

**Writing – review & editing:** Amandine Garcia, Sascha Moore Boffi, Angèle Gayet-Ageron, Nathalie Vernaz.

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
