## [Decision Letter · Decision Letter 0]

21 Aug 2019

PONE-D-19-21277

Access to unauthorized hepatitis C generics: Perception and knowledge of physicians, pharmacists, patients and non-healthcare professionals

PLOS ONE

Dear Dr Vernaz,

Thank you for submitting your manuscript to PLOS ONE. After careful consideration, we feel that it has merit but does not fully meet PLOS ONE’s publication criteria as it currently stands. Therefore, we invite you to submit a revised version of the manuscript that addresses the points raised during the review process.

We would appreciate receiving your revised manuscript by Oct 05 2019 11:59PM. To enhance the reproducibility of your results, we recommend that if applicable you deposit your laboratory protocols in protocols.io, where a protocol can be assigned its own identifier (DOI) such that it can be cited independently in the future. For instructions see: http://journals.plos.org/plosone/s/submission-guidelines#loc-laboratory-protocols

We look forward to receiving your revised manuscript.

Kind regards,

Chen-Hua Liu

Academic Editor

PLOS ONE

2. We note that your supplementary file contains a copyrighted image - Dallas Buyers Club image.

All PLOS content is published under the Creative Commons Attribution License (CC BY 4.0), which means that the manuscript, images, and Supporting Information files will be freely available online, and any third party is permitted to access, download, copy, distribute, and use these materials in any way, even commercially, with proper attribution. For more information, see our copyright guidelines: http://journals.plos.org/plosone/s/licenses-and-copyright.

We require you to either (a) present written permission from the copyright holder to publish these figures specifically under the CC BY 4.0 license, or (b) remove the figure from your submission:

a.         You may seek permission from the original copyright holder of the image to publish the content specifically under the CC BY 4.0 license.

b.    If you are unable to obtain permission from the original copyright holder to publish this image under the CC BY 4.0 license or if the copyright holder’s requirements are incompatible with the CC BY 4.0 license, please either i) remove the figure or ii) supply a replacement figure that complies with the CC BY 4.0 license. Please check copyright information on all replacement figures and update the figure caption with source information. If applicable, please specify in the figure caption text when a figure is similar but not identical to the original image and is therefore for illustrative purposes only

3. Please update your ethics statement to include the information on participant anonymity and consent present in your manuscript.

5. Please include captions for your Supporting Information files at the end of your manuscript, and update any in-text citations to match accordingly. Please see our Supporting Information guidelines for more information: http://journals.plos.org/plosone/s/supporting-information

Reviewers' comments:

Reviewer's Responses to Questions

**Comments to the Author**

1. Is the manuscript technically sound, and do the data support the conclusions?

Reviewer #1: Yes

Reviewer #2: Partly

2. Has the statistical analysis been performed appropriately and rigorously? 

Reviewer #1: Yes

Reviewer #2: Yes

3. Have the authors made all data underlying the findings in their manuscript fully available?

Reviewer #1: Yes

Reviewer #2: No

4. Is the manuscript presented in an intelligible fashion and written in standard English?

Reviewer #1: Yes

Reviewer #2: No

5. Review Comments to the Author

Reviewer #1: 1. Please address the current regulations made by the Swiss government to import unauthorized generics and to the Buyer’s Club, and the rationale of allowing a maximum quantity equivalent to one month of therapy at a standard dosage from Swissmedic.

2. You may cite Liu’s (not ‘Lui’ [44]) another published paper investigating generic DAA for CHC patients on Sci Rep 2018 12;8(1):13699 for reference.

3. Add ‘analysis’ after ‘univariate’ and delete ‘was’ in the first sentence on page 13.

4. Polish the language for the 2nd sentence, 2nd paragraph of page 14 (…had the same…than…).

5. The ‘middle level’ educated participants are relatively small in number as compared with ‘high level’ persons, is this true to the general population in Geneva?

Reviewer #2: This questionnaire study conducted an online anonymous survey based on the fictitious situation of a patient diagnosed with HCV who lacked mandatory health insurance and personal financial resources. This study aimed to assess the knowledge and perceptions of stakeholders on unauthorized generic DAA for HCV. The limited knowledge of the efficacy and quality of unauthorized generic DAA emphasizes the need for clearer information on imported generics and the potential safety provided by buyers’ club schemes completing the WHO agenda to eradicate viral hepatitis by 2030.

1. The details of questionnaire are S1. Did the information of the efficacy and quality of unauthorized generic DAA and real world studies (generic DAA is well-tolerated and provides comparably high SVR12 rates for HCV infection) be introduced before the questionnaire?

2. How many the registered physicians at HUF and Lausanne CHUV? The speciality of physicians, such as hepatologist, infectionist ?

3. The manuscript needs linguistic improvement and English editing, such as some redundant sentences, type error.

6. PLOS authors have the option to publish the peer review history of their article (what does this mean?). If published, this will include your full peer review and any attached files.

Reviewer #1: No

Reviewer #2: No

---

## [Author Response · Author response to Decision Letter 0]

15 Sep 2019

Revised PONE-D-19-21277 entitled «Access to unauthorized hepatitis C generics: Perception and knowledge of physicians, pharmacists, patients and non-healthcare professionals» by Garcia A et al.

We would like to thank you and the reviewers for the careful review of our above-mentioned manuscript. We thank the reviewers and the editor for the helpful and constructive comments guiding our revision and the appreciation of the importance of our work. 

A point-by-point response to all comments regarding our submission is addressed. 

. 

A. Answers to the comments by the editor 

Editor 1. To enhance the reproducibility of your results, we recommend that if applicable you deposit your laboratory protocols in protocols.io, where a protocol can be assigned its own identifier (DOI) such that it can be cited independently in the future. For instructions see: http://journals.plos.org/plosone/s/submission-guidelines#loc-laboratory-protocols.

-We agree to the Plos One recommendation in order to enhance the reproducibility of the results and will therefore provide an excel file of primary data upon acceptance. 

Editor 2. Please ensure that your manuscript meets PLOS ONE's style requirements, including those for file naming. 

- We thank the editor for this comment. We now modified our manuscript, title author’s affiliations, main body manuscript including tables, references and supporting information in order to meet PLOS ONE's style requirements.

Former reference 32: “Gustafsson BGBWIBTBJFKGOLJEMCSITCZLL. European payer initiatives to 521 reduce prescribing costs through use of generics. GaBI Journal. 2012;1(1):1:22.” Is now reference 36 and was modified as follow: “Godman B, Wettermark B, Bishop I, Burkhardt; T, Fürst J, Garuoliene K, et al. European payer initiatives to reduce prescribing costs through use of generics. GaBI Journal. 2012;1(1):1:22.”

Editor 3. We note that your supplementary file contains a copyrighted image - Dallas Buyers Club image.

- We thank the editor for this important comment and have removed the copyrighted image from the supporting information. 

Editor 4. Please update your ethics statement to include the information on participant anonymity and consent present in your manuscript.

- We thank the editor for raising this important issue and added in the marked-up copy the following sentence in line 164, page 9: 

“Participation in our survey was voluntary. By completing the survey, all respondents implicitly consented to the use of their data in a scientific study. Because no personal information was collected, no email addresses or IP addresses were logged and all data was collected and analysed anonymously thus guaranteeing the anonymity of all respondents, this study is considered to be exempt from formal institutional review. We added the reference 29: “Article 2 para. 2 of the Federal Act on Research involving Human Beings (HRA) [cited 2019 September 10] Available from: https://www.admin.ch/opc/en/classified-compilation/20061313/index.html).”

- Editor 5. We note that you have included the phrase “data not shown” in your manuscript. Unfortunately, this does not meet our data sharing requirements. PLOS does not permit references to inaccessible data. We require that authors provide all relevant data within the paper, Supporting Information files, or in an acceptable, public repository. 

We thank the editor for this important comment. We have removed the phrase “data not shown” and linked the corresponding results to the corresponding table to the S3 Table: Univariate analysis of the supporting information. The sentence is now as follows in the marked-up copy in line 228, page 14: “In the univariate analysis (see S3Table), we found that the physician and non-healthcare professional groups were consistently associated with all four primary outcomes: compared to pharmacists, physicians and non-healthcare professional were, respectively, 3.40 and 5.27 times more willing to agree regarding the efficacy of imported generics, 3.36 and 4.52 times more willing to agree about their quality, 2.40 and 2.75 times more willing to agree about the need for quality control, and 3.16 and 3.47 times more willing to agree about the need to financially support this initiative.”

Editor 6. Please include captions for your Supporting Information files at the end of your manuscript, and update any in-text citations to match accordingly. 

- This is now addressed in in the marked-up copy line 598, page 33 and updated throughout the manuscript:

Supporting Information 

S1 Text: Fictitious case. A 28-year-old woman diagnosed with hepatitis C who has no access to insurance and does not have the financial resources necessary to personally cover the costs.

S1 Table: Survey. The survey includes 21 questions about the participants’ knowledge and perceptions regarding imported generics and the related risks.

S2 Table: Survey results-

S3 Table: Univariate analysis

B. Answers to the comments by the reviewers #1

Reviewer #1: 1. Please address the current regulations made by the Swiss government to import unauthorized generics and to the Buyer’s Club, and the rationale of allowing a maximum quantity equivalent to one month of therapy at a standard dosage from Swissmedic.

- We thank the reviewer or this important comment and modified line 89, page 6 added in the marked-up copy as follows:

“Swiss law allows any individual to import ready-to-use unauthorized medicines for their personal use provided the imported quantity is small and is for their personal consumption.(16, 17) We specified the reference 16 as follow: “Article 20 para. 2 a) of the Federal Act on Medicinal Products and Medical Devices (TPA) [cited 2019 September 10, 2019]. Available from:https://www.admin.ch/opc/en/classified-compilation/20002716/index.html.” and added the reference 17: “Article 48 of the Ordinance on licensing of medicinal products (AMBV). [cited 2019 September 10]. Available from: https://www.admin.ch/opc/en/classified-compilation/20180857/index.html.”

We also modified in line 91, page 6 in the marked-up copy as follows:“With the aim of protecting the swiss market and patients from counterfeit or substandard medicines, the regulating authority, the Swiss Agency for therapeutic products or Swissmedic, limits personal imports to a maximum quantity equivalent to one month of therapy at a standard dosage as indicated, for example, in the patient information leaflet.(18)” We added the reference 18: “Swissmedic. Guideline on medicines and the Internet. [cited 2019 February 17]. Available from: https://www.swissmedic.ch/swissmedic/en/home/humanarzneimittel/market-surveillance/medicinal-products-from-the-internet/leitfaden-arzneimittel-aus-dem-internet.html”

Reviewer #1:2. You may cite Liu’s (not ‘Lui’ [44]) another published paper investigating generic DAA for CHC patients on Sci Rep 2018 12;8(1):13699 for reference.

- We thank the reviewer for his important comment and modified the phrase in line 325, page 19 “Liu and colleagues”. We also added the following published paper investigating generic DDA for chronic HCV-infected patients “Liu CH, Huang YJ, Yang SS, Chang CH, Yang SS, Sun HY, et al. Generic sofosbuvir-based interferon-free direct acting antiviral agents for patients with chronic hepatitis C virus infection: a real-world multicenter observational study. Sci Rep. 2018;8(1):13699”, reference 37 at line 297 page 18 and line 319 at page 19.

We also added the reference 19: “Freeman JA, Hill A. The use of generic medications for hepatitis C. Liver Int. 2016;36(7):929-32.” In lines 105 and 106 page 9 and line 323 page 19.

Reviewer #1:3. Add ‘analysis’ after ‘univariate’ and delete ‘was’ in the first sentence on page 13.

- We thank the reviewer for his comment and modified the phrase as follow in line 228, page 14: “In the univariate analysis (see S3 Table),”

Reviewer #1:4. Polish the language for the 2nd sentence, 2nd paragraph of page 14 (…had the same…than…).

- We thank the reviewer for his comment and have sent our manuscript and supporting information to the American Journal Experts for further editing. Line 249, page 15 is now:

“Medical doctors, patients and non-healthcare professionals had all a higher agreement with the statement that Indian/Egyptian generics had the same efficacy as Swiss drugs.”

Reviewer #1:5. The ‘middle level’ educated participants are relatively small in number as compared with ‘high level’ persons, is this true to the general population in Geneva?

- We thank the reviewer for this important comment. Due to time constraints and the study horizon, we did not conduct a survey of the general population. We cannot answer this question. This is a limitation and as mentioned an interesting future analysis in line 393, page 22: “to establish to what extent the general population is familiar with these concepts and what measures could be put in place to increase their knowledge and understanding.”

C. Answers to the comments by the reviewers #2

- We thank the reviewer #2 for his constructive comments and and the appreciation of the importance of our work.

Reviewer #2:1 The details of questionnaire are S1. Did the information of the efficacy and quality of unauthorized generic DAA and real world studies (generic DAA is well-tolerated and provides comparably high SVR12 rates for HCV infection) be introduced before the questionnaire?

- We didn’t include any information that might have introduced bias to the survey. The only information the respondents had before completing the survey is addressed in the supporting information. Please note the same survey was addressed to different stakeholders including non-healthcare professionals and meant to be comprehensive and simple. 

Reviewer #2:2. How many the registered physicians at HUF and Lausanne CHUV? The speciality of physicians, such as hepatologist, infectionist ?

- The are around 15 hepatologists out of 1370 medical doctors at HUG and around 10 out of around 500 medical doctor who received the survey at CHUV.

3. The manuscript needs linguistic improvement and English editing, such as some redundant sentences, type error.

- We thank the reviewer for this comment and have sent our manuscript and supporting information to the American Journal Experts for further editing. 

Two redundant sentences were removed:

in lines 79 to 82, page 5: “In this context, prices are reduced significantly in response to intense competition, enabling medicine to be affordable, even for life-saving drugs such as those used to treat the HIV and HCV epidemics” 

and in lines 292 and 293, page 17: “However, these drugs represent an important step forward in terms of public health and sustainable financing.“

---

## [Decision Letter · Decision Letter 1]

26 Sep 2019

Access to unauthorized hepatitis C generics: Perception and knowledge of physicians, pharmacists, patients and non-healthcare professionals

PONE-D-19-21277R1

Dear Dr. Vernaz,

We are pleased to inform you that your manuscript has been judged scientifically suitable for publication and will be formally accepted for publication once it complies with all outstanding technical requirements.

With kind regards,

Chen-Hua Liu

Academic Editor

PLOS ONE

Reviewers' comments:

Reviewer's Responses to Questions

**Comments to the Author**

1. If the authors have adequately addressed your comments raised in a previous round of review and you feel that this manuscript is now acceptable for publication, you may indicate that here to bypass the “Comments to the Author” section, enter your conflict of interest statement in the “Confidential to Editor” section, and submit your "Accept" recommendation.

Reviewer #1: All comments have been addressed

Reviewer #2: All comments have been addressed

2. Is the manuscript technically sound, and do the data support the conclusions?

Reviewer #1: (No Response)

Reviewer #2: Yes

3. Has the statistical analysis been performed appropriately and rigorously? 

Reviewer #1: (No Response)

Reviewer #2: Yes

4. Have the authors made all data underlying the findings in their manuscript fully available?

Reviewer #1: (No Response)

Reviewer #2: Yes

5. Is the manuscript presented in an intelligible fashion and written in standard English?

Reviewer #1: (No Response)

Reviewer #2: Yes

6. Review Comments to the Author

Reviewer #1: (No Response)

Reviewer #2: Authors had responses to the questions properly. I have no other comments for the authors to publish this article.

7. PLOS authors have the option to publish the peer review history of their article (what does this mean?). If published, this will include your full peer review and any attached files.

Reviewer #1: No

Reviewer #2: No

---

## [Editor Report · Acceptance letter]

2 Oct 2019

PONE-D-19-21277R1 

Access to unauthorized hepatitis C generics: Perception and knowledge of physicians, pharmacists, patients and non-healthcare professionals 

Dear Dr. Vernaz:

I am pleased to inform you that your manuscript has been deemed suitable for publication in PLOS ONE. Congratulations! Your manuscript is now with our production department. 

With kind regards,

on behalf of

Dr. Chen-Hua Liu 

Academic Editor

PLOS ONE